# Prolonged Collagen Peptide Supplementation and Resistance Exercise Training Affects Body Composition in Recreationally Active Men

**DOI:** 10.3390/nu11051154

**Published:** 2019-05-23

**Authors:** Marius Kirmse, Vanessa Oertzen-Hagemann, Markus de Marées, Wilhelm Bloch, Petra Platen

**Affiliations:** 1Department of Sports Medicine and Sports Nutrition, Ruhr University Bochum, Gesundheitscampus Nord 10, 44801 Bochum, Germany; vanessa.oertzen-hagemann@rub.de (V.O.-H.); markus.demarees@rub.de (M.d.M.); petra.platen@rub.de (P.P.); 2Department of Molecular and Cellular Sports Medicine, German Sport University Cologne, Am Sportpark Müngersdorf 6, 50933 Cologne, Germany; w.bloch@dshs-koeln.de

**Keywords:** hydrolyzed collagen peptides, supplementation, resistance exercise training, cross-sectional area, recreational men

## Abstract

We aimed to determine the effects of long-term collagen peptide (CP) supplementation and resistance exercise training (RET) on body composition, strength, and muscle fiber cross-sectional area (fCSA) in recreationally active men. Fifty-seven young men were randomly and double-blinded divided into a group receiving either collagen peptides (COL, 15 g/day) or a placebo (PLA). Strength testing, bioimpedance analysis, and muscle biopsies were used prior to and after an RET intervention. Food record protocols were performed during the RET intervention. The groups trained three times a week for 12 weeks. Baseline parameters showed no differences between groups, and the external training load and dietary food intake were also similar. COL showed a significant increase in fat-free mass (FFM) compared with the placebo group (*p* < 0.05). Body fat mass (BFM) was unchanged in COL, whereas a significant increase in BFM was observed in PLA. Both groups showed significant increases in all strength tests, with a trend for a slightly more pronounced effect in COL. The fCSA of type II muscle fibers increased significantly in both groups without differences between the two groups. We firstly demonstrated improved body composition in healthy, recreationally active men subsequent to prolonged CP supplementation in combination with RET. As the observed increase in FFM was not reflected in differences in fCSA hypertrophy between groups, we assume enhanced passive connective tissue adaptations in COL due to CP intake.

## 1. Introduction

The use of various protein supplements in combination with exercise regimens, such as resistance exercise training (RET), is well established in elite athletes as well as in recreationally active individuals. The effects of additional protein intake are widely known to augment muscle protein biosynthesis (MPS) [1], promote higher muscle fiber cross-sectional area (fCSA) and strength enhancement [2,3], and benefit body composition by increasing fat-free mass (FFM) and decreasing body fat mass (BFM) [2]. Furthermore, postexercise recovery can be positively affected by different protein intake strategies, implying that additional protein facilitates muscle repair, immune function, and muscle remodeling [4]. These effects have been predominately described for essential/branched-chained amino acids such as leucine, which is associated with improved muscle cell metabolism as a result of triggering the mammalian target of rapamycin (mTOR) pathway [5].

Besides contractile muscle fiber adaptations, passive tissue components are also able to adapt to mechanical loads [6,7,8]. The muscle and tendon cells surrounding connective and adhesive tissue—the extracellular matrix (ECM)—may be important factors for contractile force development [9]. The regulatory mechanisms of passive tissue protein biosynthesis, triggered by stretching the tissue or muscular activity, were well described by Kjær [10], but are only partly understood. However, it is unclear whether a specific protein supplementation is able to enhance the adaption of the mentioned passive tissue components. The primary structural protein of connective tissues is collagen [4]. Approximately 25–30% of total protein mass in human bodies is collagen [11], and it is ubiquitous within the ECM tissue [12]. The positive effects of collagen peptide (CP) supplementation in wound healing, diverse reduced joint and tendon pains, and increased subjective ankle stability in injured athletes and patients have been previously described [13,14,15,16,17,18]. CP supplementation in combination with specific exercise-based rehabilitation is very likely to support recovery [19]. However, there is a lack of information regarding the body composition or strength benefits of CP supplementation in healthy populations engaged in a regular RET regime. Up to now, there has been no consideration given to the potential ability of specific collagen intake in combination with RET to increase contractile muscle tissue, and it is probably because of its low leucine content [11].

There is evidence for acute augmented collagen synthesis after CP intake followed by mechanical loads in the form of lower body plyometric exercise [20]. Nevertheless, little is known concerning CP supplementation combined with a prolonged RET regime, and in this context, few studies have looked at strength and body composition in healthy individuals. It is assumed that the satellite cells of contractile muscle fiber cells are crucial for muscle repair and muscle fiber hypertrophy and can interact with myogenic progenitor cells in the surrounding ECM [21]. This cross-talk could lead to either stimulation of ECM synthesis by satellite cells [21] or stimulation of satellite cell activation after muscle injury by fibroblasts [22,23]. Regarding this, a higher quantity of ECM, including higher numbers of fibroblasts, could have a regulatory influence on satellite cell activity and, therefore, muscle repair and muscle fiber hypertrophy. However, this assumption is based on research of in vitro models. Yet, the question remains whether prolonged CP intake in combination with RET can increase the quantity of passive and connective tissue cells and, moreover, whether this supplement somehow affects the contractile muscle fiber size of healthy individuals.

Zdzieblik et al. [11] were the first investigators to show a positive effect of CP supplementation in combination with RET on strength, body composition (as increased FFM), and motor control in sarcopenic elderly individuals compared with the placebo group. These results were based on noninvasive methods, and no information can be given about the composition of the FFM. Therefore, it is unclear whether the increase in FFM and its relation to greater strength enhancement is related to higher contractile cell mass or a higher volume of connective tissue or both.

Accordingly, the aim of this study was to re-examine the results of Zdzieblik et al. [11] in a younger cohort experienced in resistance training as well as to gather new information about the composition of the FFM by adding muscle biopsies to methodology to quantify isolated contractile muscle fiber hypertrophy.

## 2. Materials and Methods

This investigation was designed to determine the effects on strength, body composition, and muscle fiber characteristics of CP hydrolysate supplementation in combination with a 12-week RET program. Therefore, a randomized, double-blinded, placebo-controlled design was implemented. All included participants completed the training period for a minimum of 32 training sessions, as well as the pre- and post-testing procedure. On day 1 of testing, our subjects visited the laboratory for assessment of their body composition via biometrical impedance analysis, muscle thickness via ultrasound measurement, and dynamic maximal strength via the one-repetition maximum method (1RM) in four barbell exercises. Three days later, unilateral leg extension (Leg-Ex) maximal voluntary isometric contraction (MViC) was performed on a dynamometer on day 2 of testing. Day 3 of testing consisted of muscle tissue sampling via muscle biopsy (Figure 1). The study was designed according to the adaptive design method [24]. Data and possible differences between groups were collected in the first phase (phase 1) in order to determine the sample size for the second phase (phase 2). The results and data of the two phases were merged. Thus, in the first phase of the study, 25 subjects who successfully completed the study were included for statistical analysis (age = 24 ± 3 years; height = 1.85 ± 0.05 m; weight = 79.7 ± 5.6 kg). Leg-Ex MViC was used as a predictor of potential differences in strength between groups to calculate the sample size for the second phase using G-Power [25]. With an assumed α-error of 0.05 and an assumed power of 0.8, a group size of 28 subjects per group was calculated. Assuming that the same effect sizes were expected for an identical study design, the subjects who were already included were subtracted from the calculated 28. Thus, a size of 16 subjects per group was calculated for phase 2 of the study. A total of 40 subjects were recruited, and 32 of the 40 completed the study successfully (age = 24 ± 2 years; height = 1.83 ± 0.07 m; weight = 78.0 ± 8.5 kg). Both phases applied the same methodology and main factors, including the investigators, training, and testing equipment. The process of testing on days 1 and 2 was identical between the pre- and post-testing procedure, as well as between phase 1 and 2. Muscle biopsies were only performed in pre- and post-testing procedure of phase 1 and were not repeated in phase 2.

### 2.1. Subjects

Sixty-eight subjects were recruited for this investigation. However, only 57 subjects met the inclusion criteria (completion of pre- and post-testing, no injuries during intervention, at least 32 training sessions) and were included for statistical analysis (mean ± SD: age = 24 ± 3 years; height = 1.84 ± 0.06 m; body mass = 78.8 ± 7.4 kg). Participants were moderately trained (nonstructured, consistent resistance training and no protein supplementation 6 months before starting the approach) and included on the basis of their ability to execute a barbell squat equivalent to 100% body weight at least once with the correct technique. Subjects were randomly and double-blinded assigned to either a collagen supplementation treatment group (COL, *n* = 29; age = 24 ± 2 years, height = 1.84 ± 0.07 m, body mass = 79.3 ± 8.4 kg) or a placebo supplementation control group (PLA, *n* = 28; age= 24 ± 3 years, height = 1.84 ± 0.06 m, body mass = 78.2 ± 6.3 kg). In advance, all subjects were informed verbally and in writing of the procedure and purpose of this investigation and of possible risks. Subjects were instructed to generally maintain their daily lifestyle habits, including sports activities and their habitual dietary intake, throughout the whole trial. The study was conducted in accordance with the Declaration of Helsinki, and the protocol was approved by the local Ethics Committee of the Faculty of Sport Science of Ruhr University of Bochum (EKS V 01/2016).

The study was approved by the local Ethics Committee of the Faculty of Sport Science of Ruhr University of Bochum and was conducted in accordance with the Declaration of Helsinki.

### 2.2. Resistance Training Protocol and External Load Calculation

Full-body RET with a barbell was completed in a 12-week training program. The protocol was performed by all subjects and consisted of three training sessions per week for a total of 36 sessions. In the first week of training, the subjects visited our laboratory three times for their first training experiences. The external load did not exceed 50% of 1RM in each set and exercise. In this way, the participants were familiarized with the training procedure. Furthermore, no other sports activities were allowed during the first week, so each participant started with a relatively equal preload as they entered the 12 weeks of training. After being familiarized with the training procedure, our participants started with 70% training loads and supplementation in their 4th training session. During each session, the subjects performed a standardized bodyweight warm-up. Then, the barbell exercises known as the squat (SQ), bench press (BP), deadlift (DL), and bent-over row (BR), and an additional knee extension (KE) exercise with a device, were performed in a randomized order in each training session. Each exercise started with a warm-up set of 10 repetitions with 50% of the determined 1RM followed by 3 sets of 10 repetitions with 70% of the determined 1RM. Every set was separated by 2 min of rest, just as each exercise was separated by 2 min to change weights and positions. This training protocol was designed to achieve local fatigue in each individual in each exercise and respective muscle group in the last set, i.e., 70% of 1RM. Thus, if all repetitions were completed successfully with correct technical performance, the training weight was systematically increased by 2.5 kg in BP, BR, and KE and by 5.0 kg in SQ and DL. If the exercise technique was inadequate, the weights were not increased. It was incumbent on the investigator to prevent negative health consequences using subjective estimation. If the second training set (70% 1RM) was not completed successfully, the training weight was decreased in each exercise, respectively. If pain occurred during or immediately after an exercise, the training weight was reduced for the next session. Participants were excluded from the study if they took part in fewer than 32 training sessions. The external load is defined as the product of weight and repetition in each exercise and training day, respectively [26]. To analyze individual changes in the external loads during the training period, the 4th and the 32nd training sessions were used to calculate the pre- and post-external load in each exercise.

### 2.3. Dietary Protein Supplementation

Throughout the 12-week training intervention period, subjects were given their supplements in a double-blinded manner, for a daily intake. The packages contained either 15 g of collagen (COL) or 15 g of placebo (PLA) in powder form. The supplement was dissolved in a minimum of 250 mL water and taken immediately after training on training days. In the following hour, no other intake was allowed except water. On training-free days, the subjects were instructed to ingest the supplement approximately 24 h after the previous ingestion. The supplement of COL (Bodybalance^TM^ [11]) was provided by GELITA AG (Eberbach, Germany). The placebo contained a noncaloric silicon dioxide (Sipernat 350, Evonik Industries, Essen, Germany) that does not provide energy.

### 2.4. Dietary Intake Recording

To record dietary intake habits and get an overview of total energy intake and macronutrient distribution, a self-reported protocol was assigned in the middle of the training period. This protocol includes two weekdays and one weekend day. Subjects had to note the exact weight of food contents. With this information, the total caloric intake and the total protein, carbohydrate, fat, and relative protein intake on these three days were calculated. Subjects were encouraged to maintain their normal nutritional behavior. This protocol was analyzed with PRODI^®^ 6.8 (Nutri-Science GmbH, Freiburg, Germany).

### 2.5. Body Composition Testing

On the first pre- and post-testing day, body composition was tested using a biometrical impedance analysis system (BIA; InBody 770, JP Global Markets GmbH, Eschborn, Germany) to determine total body weight (BW), fat-free body mass (FFM), and body fat mass (BFM) [27,28]. Following an overnight fast, subjects visited the laboratory. Participants were instructed not to perform any sports for at least 48 h before the day of testing. Under standardized conditions, composition testing was performed twice on each testing day, and the average of both tests was calculated for statistical analysis.

### 2.6. Ultrasound and Anthropometric Measurement

Muscle thickness was measured directly after body composition testing using a b-mode ultrasound system (LS128 CEXT-1Z, UAB Telemed, Vilnius, Lithuania). The image position was determined by an orthogonal line passing 50% of the line from the spina iliaca anterior superior to the patella [29]. Three images in two positions were taken to assess the thickness of the m. rectus femoris (Rec) and m. vastus intermedius (Int) in the first position and the thickness of the m. vastus lateralis (Lat) in the second position. Images were taken to display the mid part of the muscle belly and to draw a vertical line from the bone upward through the middle of the picture. Before taking a new picture, the ultrasound probe was lifted up completely. The pressure on the tissue and the verticalness to the skin was controlled manually before taking a single image. Afterward, all three images of both positions were measured using Image-J software (National Institute of Mental Health, Bethesda, Rockville, MD, USA), with a vertical line drawn radial to the bone used as a ledger line. Using this line as guidance, only the tissues between aponeuroses, which were not involved in the calculation, were measured. The average of the muscle thickness measurements was taken for statistical analysis. The spot of the mentioned orthogonal line was also used for measuring leg circumference (LC) with a standard measuring tape.

### 2.7. Strength Testing

One-repetition maximum (1RM) was tested by the method described by Kreamer et al. [30] in order to obtain 1RM for SQ, BP, DL, and BR. Therefore, a warm-up set of 10 repetitions with 50% of the estimated 1RM was performed, followed by another set of 5 repetitions with 80% of the estimated 1RM. Subsequently, the exercise load was systematically increased over 3–5 trials that were separated by 3–4 min of rest for 1RM determination. The performed 1RM was considered valid if the techniques explained beforehand were performed correctly in a controlled manner and without assistance. Depth- and movement-specific joint angles were monitored by the same investigators throughout every trial while testing and training. The order of exercises was recorded for post-testing. Afterward, all subjects were familiarized with the Leg-ex MViC procedure. Three days after familiarization, the previous MViC procedure was performed again on day two of testing. Participants were advised to avoid the intake of caffeine and alcohol and the performance of exhausting sports activities for 12 h and 48 h prior to MViC testing, respectively. A standardized warm-up was executed that included mobilization of the lower limbs, slow running, and bodyweight lunges. Then, three leg extension MViC’s of the right leg with 60° knee flexion were performed on a dynamometer (Isomed 2000, D&R GmbH, Hemau, Germany) with three minutes of rest between attempts. The best attempt was recorded for statistical analysis. Participants underwent the same dynamic and isometric strength testing procedure after 12 weeks of training and supplementation.

### 2.8. Muscle Biopsies and Venous Blood Sampling

Blood samples in phase 1 were collected on day 3 of post-testing. Because there was no testing day 3 in phase 2, blood samples in phase 2 were collected at the end of the training period. Muscle biopsies were performed only in phase 1 and the methods are described in the following. After a standardized breakfast, percutaneous needle muscle biopsies were taken from the m. vastus lateralis muscle of the participant’s right leg under local anesthesia (Xylocitin^®^ 2% with Epinephrine, mibe GmbH, Brehna, Germany) with a 5 mm Bergstrøm needle, as described before [31]. Immediately after collecting muscle tissue, the whole sample was weighed and cut into a coherent piece of approximately 30 mg for histochemical analysis. Directly after, this sample was mounted in a Tissue-Tek^®^ O.C.T.™ Compound (Sakura Finetek Europe B.V., Alphen aan den Rijn, Netherlands) and frozen in cooled liquid nitrogen. Thereafter, subjects performed typical training as described before, with the respective supplement intake after training and a training intensity of 70% of the determined 1RM. Two hours after training and supplement intake, a second blood sample was taken to assess the ability to utilize collagen’s typical amino acids, such as Hydroxyproline [32]. This was followed by a second biopsy from the same muscle and leg, 3 cm proximal to the previous spot, for analyzing the acute effects of CP supplementation and training. This data will be shown in further publications. The procedure was identical for both the pre- and post-testing days. For this study, only the first (pretraining) biopsies were taken for muscle fiber distribution and thickness analysis.

### 2.9. Immunohistochemical Stains of Skeletal Muscle

For analyzing muscle fiber distribution and contractile muscle fiber cross-sectional area (fCSA), muscle samples from phase 1 were cut into 7-µm-thick serial sections on a cryostat microtome (Leica CM 3050 S, Leica Biosystems, Nussloch, Germany). For muscle fiber typing, the cryosections were stained using an immunohistochemical protocol. In short, the sections were incubated overnight with primary antibodies against MHY7 (A4.951, monoclonal, mouse, DSHB, Iowa City, IA, USA) to visualize type I fiber cells and Laminin (polyclonal, rabbit, Sigma-Aldrich, St. Louis, MO, USA) to visualize cell membranes. Subsequently, goat anti-mouse and goat anti-rabbit secondary antibodies (Dako, Agilent Pathology Solutions, Santa Clara, CA, USA) were applied. After that, horseradish peroxidase (Sigma-Aldrich, St. Louis, MO, USA) was added, and 60 min later, Diaminobenzidin (Sigma-Aldrich, St. Louis, MO, USA) was applied. Between all steps, cryosections were washed using Tris-Buffered Saline (Merck KGaA, Darmstadt, Germany). With this method, type I muscle fibers were differentiated from type II to get an initial idea about changes in fiber distribution and fCSA. Images were captured at 10× magnification with a light microscope (Leica Orthoplan, Leica Microsystems GmbH, Wetzlar, Germany) using Irfanview software (Irfan Skiljan, Wiener Neustadt, Austria). The respective fiber type percentage is defined as the total fiber number of one fiber type divided by the number of all fibers in this cryosection and multiplied by 100. Fibers and fiber percentage were counted and calculated, respectively, for each subject and section. To determine the fCSA, each myofiber was traced along its laminin-stained line. The measured perimeter and area were used to calculate a roundness factor by a known formula [33]:Roundness = perimeter2/4π × area

Only the fibers that did not exceed a calculated value of 1.639 (perfect circle = 1.0; pentagon = 1.1639; square = 1.266; equilateral triangle = 1.639) were included in statistical analysis. Distribution and fCSA were measured using Image-J software (National Institute of Mental Health, Bethesda, Rockville, MD, USA). All measured fibers were averaged for each fiber type, subject, and biopsy sample.

### 2.10. Subjective Perception

Before each testing session, subjects were asked to use a visual analog scale (delayed onset of muscle soreness, DOMS) to identify muscular pain as a predictor of prefatigue [34]. Immediately after the completion of each training exercise and after each training session, the participant’s perceived exertion was recorded using the CR-10 RPE (rate of perceived exertion) scale [35] to gather information about the perceived internal load [26].

### 2.11. Statistical Analysis

All data were analyzed by two-way analysis of variance (ANOVA) with repeated measurements (time × group) using IBM SPSS Statistics version 23 (SPSS, Inc., Chicago, IL, USA). If ANOVA revealed a significant interaction effect (time × group: *p* ≤ 0.05) or a trend, post hoc analysis tests were performed (paired *t*-tests and unpaired *t*-tests for pre- and post-values) using Bonferroni correction (to *p* ≤ 0.0125). Further, partial eta squared (*η_p_*^2^) calculation was performed to demonstrate the effect size of significant interaction effects as well as for trends of interaction effects (time × group: *p* ≤ 0.05). If ANOVA revealed a significant main effect (time: *p* ≤ 0.05), the differences between pre- and post-values of the whole cohort were analyzed using a paired *t*-test. The level of statistical significance was set at *p* ≤ 0.05.

## 3. Results

Because of injuries from other activities or the completion of fewer than 32 training sessions, 11 subjects were excluded from the study. Fifty-seven participants who completed a minimum of 32 training sessions were included for statistical analysis. Because of individual health discomforts, such as acute lower back pain during post-testing or inadequate documentation of the training diary or food protocols, the number of included subjects also differs among parameters. At the baseline measurement, no differences were found in any of the measured parameters between groups (*p* > 0.05). In the following, the results represent the merged data of all included participants of phase 1 and 2. Only the last section, the biopsy data, shows the data of phase 1 solely.

### 3.1. External Load, Ultrasound, Leg Circumference, and Subjective Perception

No group interaction effect was found in any of the parameters discussed in this section (group × time: *p* > 0.05). The data of the external load of each exercise, the muscle thickness measured via ultrasound in each muscle, and the leg circumferences showed main effects with an increase in each parameter in the whole cohort from pre to post (time: *p* < 0.05; Table 1). No significant differences were found in muscular pain on testing days between groups (DOMS, *p* > 0.05), and no differences between groups were found in the responses to the RPE scale (*p* > 0.05) for each exercise within the training period.

### 3.2. Body Composition

Data from before and after the 12-week training (i.e., pre- and post-testing, respectively) are shown in Table 1. BW showed only a main effect (time: *p* < 0.001) with no interaction effect (time × group: *p* = 0.314), indicating an increase in the BW of the whole cohort. A significant interaction effect was found in FFM (time × group: *p* = 0.002, *η_p_*^2^ = 0.163). Post hoc test showed no differences between groups at post-testing values (*p* = 0.227). However, a significant increase in FFM from pre- to post-testing within groups was only found in COL (*p* < 0.001), with no statistical enhancement in PLA (*p* = 0.018). BFM also revealed a significant interaction effect (time × group: *p* = 0.027, *η_p_*^2^ = 0.085). Post hoc test showed no differences between groups at post-testing values (*p* = 0.672). Further, a significant increase in BFM in PLA (*p* = 0.003) was found with no change in BFM in COL (*p* = 0.806) from pre to post-testing within groups.

### 3.3. Strength Testing

After 12 weeks of RET and supplementation, a main effect was found in single leg extension MViC in both groups (time: *p* < 0.001), indicating higher isometric strength of the whole cohort after the intervention without an interaction effect (time × group: *p* = 0.477). Also, the measured 1RM showed a main effect for increased strength over time (time: *p* < 0.001, respectively) with no interaction effect for the exercises DL (time × group: *p* = 0.576), BP (time × group: *p* = 0.474), and BR (time × group: *p* = 0.768). SQ also showed a main effect for higher strength after the training and supplementation (time: *p* < 0.001), with a trend for an interaction effect (time × group: *p* = 0.054, *η_p_*^2^ = 0.071). Post hoc test showed no differences between groups at SQ post-testing values (*p* = 0.180). Paired *t*-tests showed that COL (*p* < 0.001) and PLA (*p* < 0.001) increased strength from pre to post-testing in SQ. The data are shown in Table 1 and Figure 2.

### 3.4. Dietary Intake Recording

After analyzing the dietary intake recordings, no statistical differences were found between groups (*p* > 0.05; Table 1). In addition, we analyzed the total calories of ingested protein per day (excluding the daily supplement) to reveal prior differences in groups. Thus, our data showed no differences between groups, excluding the daily supplement intake (total protein intake, excluding the supplement: COL: 571.5 ± 139.2 kcal/day; PLA: 556.0 ± 167.2 kcal/day, *p* > 0.711).

### 3.5. Venous Blood Sampling

Analysis of hydroxyproline from post-testing venous blood samples showed an increase in hydroxyproline levels in each subject of COL after the supplement ingestion (before ingestion: 31.1 ± 16.4 µmol/L, after ingestion: 82.0 ± 26.5 µmol/L, *p* < 0.001) and no change in the subjects of PLA (before ingestion: 14.7 ± 6.4 µmol/L, after ingestion: 13.8 ± 8.4 µmol/L, *p* = 0.546). We also detected a significant group difference in the blood samples before SUPP intake, indicating a higher level of Hydroxyproline in COL before the ingestion of the supplement on the post-testing day (COL: 31.1 ± 16.4 mL; PLA: 14.7 ± 6.4 mL, *p* < 0.001).

### 3.6. Muscle Biopsy

Not every sample was a connected piece of tissue of acceptable quality, so some could not be cut into cross-sections. Therefore, samples from 21 participants of phase 1 (10 COL, 11 PLA) were included in the histochemical analysis. A total of 8.453 cells were measured and analyzed for fiber type distribution and muscle fiber fCSA. No changes in fiber type distribution were found from pre- to post-biopsy between and within groups (Table 2). Muscle fiber CSA increased in type II (time: *p* < 0.001) and showed a trend for higher fCSA in type I muscle fibers (time: *p* = 0.099) of the whole cohort, without an interaction effect, respectively (Table 1). Because of the clear and unambiguous results (no group × time interaction effect) for all histochemical parameters in phase 1 and for ethical issues, muscle tissue sampling was not repeated in phase 2. Results are shown in Table 2.

## 4. Discussion

To our knowledge, this is the first study to investigate the effects of hydrolyzed collagen peptide supplementation in combination with RET on a cohort of recreationally active men. The aim of this study was to re-examine the results of Zdzieblik et al. [11] in a younger cohort and differentiate possible FFM changes by looking at the cross-sectional area of contractile muscle fiber cells (fCSA) after 12 weeks of RET and CP supplementation. The main result of this investigation is that a significant increase in FFM was observed for COL compared with PLA, with no differences in fCSA hypertrophy between groups. Both groups demonstrated increased strength to the same extent, with a tendency for a slightly higher increase in one out of four barbell exercises in COL.

Our groups showed high homogeneity as no group differences were found in any parameter for the baseline measurement, muscle soreness on testing days, daily energy and macronutrient intake, subjective perception, and external loads during the training period. This was highly required to compare our groups and attribute possible differences to the CP supplement. Nevertheless, group heterogeneity cannot be completely excluded because of missing data on participants’ previous experiences. No consistent training prior to the study and the correct execution of a barbell back squat with 100% of the subject’s body weight were our only inclusion criteria; therefore, this limitation must be considered.

Zdzieblik et al. [11] found higher FFM in their CP supplementation group compared with controls in a men’s cohort. This has been confirmed in another recent study that shows the same positive changes in FFM of their CP supplementing group compared to their placebo group in a women’s cohort [36]. The FFM includes passive as well as active contractile tissues [10,37]. Hence, muscle biopsies and ultrasound measurements were additionally added to our methodology to quantify and separate the active skeletal muscle cells to get a general idea about the composition of FFM for further interpretation. Notwithstanding that the ultrasound measurement denotes no gold-standard in the assessment of muscle thickness, the correlation to MRT analysis was found to be significant [29] and the variation coefficient as well as the intraclass correlation coefficient indicate high reliability as already examined [38,39]. Our findings on body composition parameters are partly in line with already published results [11,36]. As already shown in older sarcopenic men and premenopausal women, we also measured a significant increase in FFM in COL compared with PLA. According to the analysis of fCSA and ultrasound images, fCSA and muscle thickness increased for the whole cohort, indicating that our RET affected the muscle size to the same degree in COL and PLA, and no differences in hypertrophy were observed between our groups. The greater hypertrophy of the type II fibers is likely to be because of the high external load resistance training protocol (70% 1RM) and their greater capacity to increase in size compared with type I fibers [40]. Because fCSA was only analyzed in Phase 1, an additional ANOVA was used to determine potential differences in FFM and ultrasound analysis between Phase 1 and Phase 2. Therefore, no difference was found in the pre- and post-testing data between Phase 1 and Phase 2 in COL and PLA. So, because of the lack of differences in the development of FFM and ultrasound analysis between phase 1 and phase 2, we assume the same development of fCSA for the respective groups in each phase. Hence, we were able to exclude higher contractile muscle cell hypertrophy in COL subjects as a factor that explained the group differences in FFM.

Consequently, our assumption is that the passive tissue components of both groups adapted to the training as a physiological reaction [12,41,42,43], but more quickly and to a higher extent in COL compared with PLA due to the specific supplementation. Recent research has already shown enhanced acute collagen synthesis with CP intake and specific exercise [20]. Additionally, we further assume that this acute enhancement results in higher passive tissue mass after 12 weeks of CP intake with exercise. However, this assumption is based on current evidence-based research and the exclusion process due to the fCSA results of this investigation; more research is needed to verify our results with additional quantification of these passive tissues.

Although the FFM enhancements were significantly higher in COL, both groups increased their body weight to the same extent. This is explained by a significant increase in the body fat mass (BFM) of PLA, whereas COL showed no difference in this variable from pre- to post-testing. This is in contrast to the study on elderly men and premenopausal women in which each group lost BFM, but a significantly higher loss was found in the CP supplementation group [11,36]. However, these subjects had approximately 30% [11] and 37% [36] body fat, and we detected 11.3 ± 4.0% body fat in our recreationally active men at the baseline. Because of the higher fitness level per se and physical activity prior to and during the current investigation, a loss of BFM was not hypothesized anyway.

Nevertheless, despite equal external loads and energy intake, the interaction effect found for FFM and BFM could indicate different energy requirements. Assuming that an adaptation of passive tissue components was enhanced by the CP supplement in COL, this could have led to higher energy requirements by the structure building state. However, the additional dose of 60 kcal per day (251 kJ) in the CP supplement for COL provides no explanation for an enhanced anabolic effect [44]. On the basis of the food record results, we assume similar habitual nutrition behavior in both COL and PLA subjects. However, a 3-day self-reported food protocol was used to represent a 3-month period. The usefulness of this protocol has been debated in recent years, and its validity can also be discussed [45]. Certainly, it has been and remains an easily accessible, useful, and common tool to get a general idea about a participant’s nutritional behaviors. Nevertheless, differences in energy intake cannot be completely excluded, and potential differences could explain higher BFM in PLA after the intervention.

Our study was based on the results of the study of Zdzieblik et al. [11]. In our healthy and younger population, we observed increased FFM, no change in BFM, and almost no difference in strength enhancement in our collagen treatment group compared with controls. As this was only partly in line with the results of Zdzieblik et al. [11], who additionally found higher increases in strength and a more pronounced loss of BFM in the collagen supplementation group compared with placebo-controlled subjects, this needs to be discussed. It is often assumed that daily protein intake in elderly populations is even less than the recommended dietary allowance (RDA) or that even more protein per day is needed to ensure age-appropriate muscular system function [46]. In a study comparing the effects of whey and fortified collagen hydrolysate on nitrogen balance in elderly subjects, no difference was found between the high- and low-quality proteins [47]. Therefore, the authors suggested that collagen-based proteins can be seen as an appropriate supplement for elderly individuals and considered equal to whey protein. Furthermore, the loss in body weight of the whey protein group could not be explained by a loss of fat mass, indicating a loss of FFM, which was not found in the collagen group [47]. It therefore could be suspected that the greater increase in FFM after collagen intake and RET in the study of Zdzieblik et al. [11] is the result of enhanced connective tissue adaptation as well as pronounced hypertrophy of contractile muscle cells; these mechanisms would explain the significant differences in strength enhancement. However, in general, the results of Zdzieblik et al. [11] conflict with the results of a systematic review in which no further enhancement was found by an additional protein supplement in combination with exercise compared with exercise alone [48]. However, only studies with high-quality supplements that are rich in essential amino acids (EAA) or EAAs themselves were included in this review, and CP supplementation was not considered. Therefore, collagen-based proteins could play a more crucial role in the elderly population than the other mentioned protein supplements. Interestingly, Jendricke et al. [36] showed the same results in premenopausal women as previously shown for the elderly men after prolonged CP intake. Due to the missing data about absolute macronutrient intake, it could be speculated that the daily protein intake was insufficient within a strength training regime. This would strengthen the assumption that a prolonged intake of CP may be equivalent to other protein intake strategies during the consumption of low-protein diets [47].

In our younger subjects, in both groups, we found an average protein intake of 1.8 ± 0.5 g/kg-bw/day, which is a sufficient amount to ensure adaptations in an RET regime following already published recommendations [1] and more than double the RDA of 0.8 g/kg-bw/day for the general population. The high and equivalent amount of daily protein intake by our groups could explain the same levels of hypertrophy, as indicated by equivalent fCSA increases, which concurrently explains the similar strength levels. However, CP intake still had an impact on FFM in our COL individuals, who also had a tendency toward a slightly higher strength increase in squat 1RM. However, this impact does not seem to affect the strength enhancement of younger healthy men to the level it affected the elderly population. This discrepancy is probably due to a daily protein supply that was already sufficient and the absence of an additive effect on contractile muscle fiber adaptation.

The assumed adaptation of the passive connective tissue in COL seems to have had no significant effect on strength enhancement. It is conceivable that the training variables and strength measurements were not representative of the effects. Assuming that COL subjects increased their connective and passive tissues in the muscle–tendon system, this could possibly have an impact on fast and reactive movements, such as sprints or jumps, but not in slow movements such as those in our 10-repetition training and the 1RM and MViC testing methods. There is current evidence that supports that CP consumption before and after a large number of exhausting drop jumps leads to better recovery [49]. Therefore, the CP treatment group returned more quickly to their previously determined maximum heights in the countermovement jump after the muscle damage protocol. This underpins the assumption that CP supplementation could be promoting recovery and adaptation in fast movement patterns.

Oesser et al. [50] found increased rates of proline—a typical collagen amino acid—in the blood plasma of mice even 48 h after ingestion. We analyzed blood samples after 12 weeks of daily supplement intake that were collected approximately 24 h after the most recent ingestion. Therefore, this could explain the higher hydroxyproline values even before the supplement ingestion in COL. This signifies the availability of collagen-like amino acids in human blood for at least 24 h after the most recent consumption in a long-term daily intake strategy. This can be considered when creating supplementation strategies or in further research on collagen intake. Accordingly, regarding long-term supplementation, a smaller dose per day or intake of CP every second day could be enough to maintain the stable availability of specific amino acids in the blood and also be sufficient for the collagen synthesis rate enhancements observed in subjects taking the supplement [20].

## 5. Conclusions

Following RET and CP supplementation, FFM increased, while BFM remained unchanged. The changes in muscle strength did not show significant differences between RET combined with CP supplementation and RET only. However, there was a tendency for a more pronounced enhancement in one strength test in COL. In addition to the subject collective, one of the novelties of this study is the use of biopsies. As fCSA did not differ significantly between the groups, we assume that the increase in FFM after CP supplementation was not only associated with hypertrophy of contractile muscle cells but might also be caused by a higher increase in connective tissue compared with placebo-controlled subjects. Further studies should focus on the quantification of the passive connective tissues after prolonged CP intake in healthy populations. A variation of the training variables and testing procedures in terms of movements, such as introducing reactive stretching and shortening cycles, could also be considered.

## Figures and Tables

**Figure 1 nutrients-11-01154-f001:**
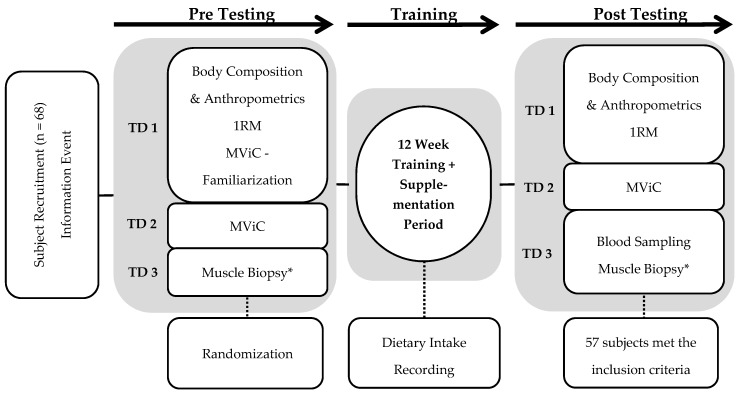
Study intervention overview: TD = testing day; 1RM = one-repetition maximum; MViC = maximal voluntary isometric contraction; * = Muscle Biopsies were only taken in Phase 1.

**Figure 2 nutrients-11-01154-f002:**
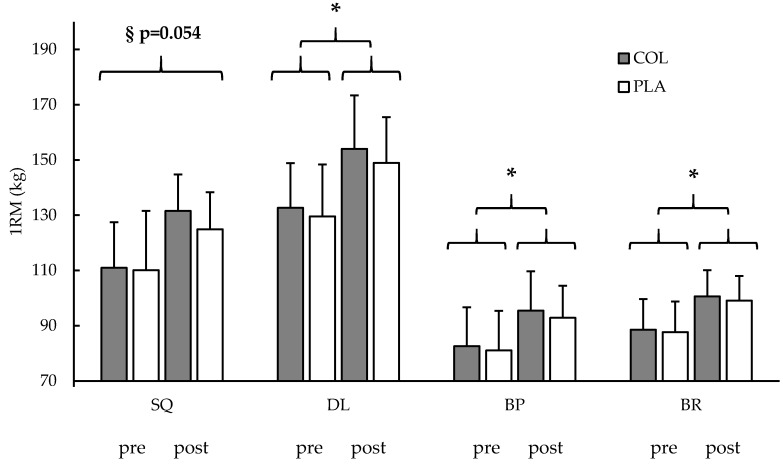
Dynamic maximal strength (1RM) before (pre) and after (post) a 12-week training period with a collagen peptide (COL, gray blocks) or a placebo (PLA, white blocks) supplementation. SQ = squat; DL = deadlift; BP = bench press; BR = bent-over row. § = trend for an interaction effect (time × group: *p* = 0.054); * = significant main effect (time: *p* < 0.05). Values are means ± SD.

**Table 1 nutrients-11-01154-t001:** Data of body composition, ultrasound, external load (from the 4th (pre) to the 32nd (post) training day), and strength of the whole cohort and of the tested groups before (pre) and after (post) a 12-week training period with a collagen peptide (COL) or a placebo (PLA) supplementation. The food protocol was performed during the 12 weeks of training for 3 days and averaged for one day. BW = body weight; BFM = body fat mass; FFM = fat-free mass; LC = leg circumference; Rec = m. rectus femoris; Int = m. vastus intermedius; Lat = m. vastus lateralis; n.s. = not significant (*p* > 0.05); * = significantly different to pre within the respective group (paired *t*-test, *p* ≤ 0.05); **^§^** = significantly different to pre of the whole cohort (ANOVA: main effect, *p* ≤ 0.05).

	Characteristics	Whole Cohort			COL			PLA			ANOVA (Time × Group)	Unpaired *t*-Test (Post)
		*n*	Pre	Post	*n*	Pre	Post	*n*	Pre	Post	*p*-Value	*p*-Value
Body Composition	BW (kg)	57	78.8 ± 7.7	80.5 ± 7.1 ^§^	29	79.3 ± 8.4	81.3 ± 8.1 *	28	78.2 ± 6.3	79.6 ± 6.0 *	n.s. 0.314	-
	BFM (kg)	57	9.0 ± 3.5	9.3 ± 3.5 ^§^	29	9.2 ± 3.8	9.2 ± 3.9 *	28	8.8 ± 3.2	9.5 ± 3.0 *	0.027	n.s. 0.672
	FFM (kg)	57	69.8 ± 6.4	71.1 ± 6.3 ^§^	29	70.1 ± 6.7	72.1 ± 6.6 *	28	69.4 ± 6.2	70.1 ± 5.9 *	0.002	n.s. 0.227
	LC (cm)	57	57.2 ± 3.3	58.4 ± 3.0 ^§^	29	57.4 ± 3.8	58.7 ± 3.5 *	28	57.1 ± 2.7	58.0 ± 2.4 *	n.s. 0.386	-
Ultra- sound	Rec (mm)	56	25.51 ± 3.25	26.08 ± 2.83 ^§^	28	25.45 ± 3.42	26.10 ± 3.09	28	25.57 ± 3.15	26.06 ± 2.60	n.s. 0.770	-
	Int (mm)	56	19.10 ± 3.52	20.57 ± 3.42 ^§^	28	19.31 ± 3.70	21.21 ± 3.89	28	18.88 ± 3.37	19.92 ± 2.78	n.s. 0.194	-
	Lat (mm)	56	26.36 ± 4.08	28.69 ± 3.96 ^§^	28	26.10 ± 3.82	28.75 ± 3.35	28	26.63 ± 4.37	28.62 ± 4.56	n.s. 0.177	-
External load	Squat (kg)	56	2735 ± 544	3280 ± 429 ^§^	29	2861 ± 463	3339 ± 457	27	2599 ± 598	3217 ± 395	n.s. 0.198	-
	Deadlift (kg)	56	3251 ± 478	3783 ± 550 ^§^	29	3338 ± 430	3836 ± 537	27	3158 ± 517	3726 ± 568	n.s. 0.503	-
	Bench press (kg)	56	2052 ± 379	2356 ± 360 ^§^	29	2069 ± 376	2380 ± 409	27	2034 ± 388	2330 ± 409	n.s. 0.803	-
	Bent-over row (kg)	56	2198 ± 360	2526 ± 347 ^§^	29	2177 ± 374	2564 ± 349	27	2177 ± 374	2564 ± 349	n.s. 0.070	-
Strength	Leg-Ex (N×m)	55	266.4 ± 46.5	291.8 ± 54.4 ^§^	28	271.5 ± 51.4	299.5 ± 61.6 *	27	261.2 ± 41.2	283.7 ± 45.6 *	n.s. 0.477	-
	Squat (kg)	53	110.5 ± 14.9	128.4 ± 18.2 ^§^	28	110.9 ± 16.5	131.5 ± 21.4 *	25	110.1 ± 13.2	124.9 ± 13.4 *	n.s. 0.054	n.s. 0.180
	Deadlift (kg)	54	131.2 ± 17.7	151.6 ± 17.8 ^§^	28	132.7 ± 16.2	154.0 ± 18.8 *	26	129.6 ± 19.4	148.9 ± 16.6 *	n.s. 0.576	-
	Bench press (kg)	56	81.8 ± 14.0	94.2 ± 13.0 ^§^	28	82.6 ± 14.0	95.4 ± 14.3 *	28	81.0 ± 14.2	92.9 ± 11.6 *	n.s. 0.474	-
	Bent-over row (kg)	56	88.1 ± 10.2	99.9 ± 10.0 ^§^	29	88.5 ± 11.0	100.6 ± 11.1 *	27	87.6 ± 9.5	99.1 ± 8.9 *	n.s. 0.768	-
Food protocol	Total Energy (kcal/day)	54	3014 ± 634		27	2958 ± 676		27	3070 ± 597		-	n.s. 0.521
	Carbohydrate (kcal/day)	54	1442 ± 367		27	1386 ± 390		27	1498 ± 341		-	n.s. 0.269
	Fat (kcal/day)	54	1001 ± 332		27	986 ± 370		27	1016 ± 396		-	n.s. 0.745
	Protein (kcal/day)	54	570 ± 153		27	585 ± 139		27	556 ± 167		-	n.s. 0.491
	Protein (g/kg/day)	54	1.77 ± 0.46		27	1.81 ± 0.42		27	1.74 ± 0.50		-	n.s. 0.610

**Table 2 nutrients-11-01154-t002:** Muscle fiber distribution (%) and muscle fiber cross-sectional area (fCSA) before (pre) and after (post) a 12-week training period with collagen peptide (COL) or placebo (PLA) supplementation. n.s. = *p* > 0.05.

	COL (*n* = 10)		PLA (*n* = 11)		ANOVA	
Variable	Pre	Post	Pre	Post	Time (*p*)	Time × Group (*p*)
Type I %	40 ± 10	37 ± 11	37 ± 13	38 ± 9	n.s. 0.692	n.s. 0.649
Type II (%)	60 ± 10	63 ± 11	63 ± 13	62 ± 9	n.s. 0.692	n.s. 0.649
Type I fCSA (µm^2^)	6455 ± 1462	6883 ± 1650	6419 ± 1094	6886 ± 1120	0.099	n.s. 0.941
Type II fCSA (µm^2^)	7258 ± 1444	8330 ± 2076	7501 ± 1604	8484 ± 1812	<0.001	n.s. 0.865

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
