# Peer review of "Prolonged Collagen Peptide Supplementation and Resistance Exercise Training Affects Body Composition in Recreationally Active Men"

_nutrients, 2019, doi:10.3390/nu11051154_

Reviewer 1 Report

Authors present a well-written study on the impact of collagen protein and resistance exercise training in active younger adults. There are a handful of items that should be addressed to strengthen the presentation of the findings as well as the readership of the paper.

The statement, ”…it is unclear whether additional protein supplementation is able to enhance the beneficial effects of RET…” is not necessarily true. It has been repeatedly reported that Leucine, BCAAs, whey protein (isolate, hydrolysate, or concentrate), and plant-based proteins are indeed ergogenic in combination with RET. This statement should be modified or removed.

The description of phase 1 and phase 2 to determine appropriate sample size is very confusing. It seems that phase 1 was exploratory and not sufficient, therefore additional subjects were recruited. This could be elaborated on briefly.

Figure 1 indicated there was blood sampling, but only during post-testing. The description of blood sampling (section 2.8) was unclear if phase 1 blood sampling was day 3 of pre- or post-testing. However, only phase 2 was collected post-testing?

Why were muscle biopsies only taken during phase 1?

Was MViC conducted on the subject’s dominant leg? Or simple on all right or all left legs, based on how the dynamometer is set up? If the latter (ex, all right legs), then leg dominance should be noted as well as any recent (within 2 years) right leg injury (ACL, meniscus, strain, etc.).

The methods should be listed in the order that they occurred, therefore, the sections after general study design should be re-ordered:

2.1-Subjects

2.2-Body Composition Testing

2.3-Ultrasound and Anthropometric Measurement

2.4-Strength Testing

2.5-Muscle Biopsies and Venous Blood Sampling

     2.5.1 Immunohistochemical Stains of Skeletal Muscle-This is not a method involving contact with the subject

2.6-Resistance Training Protocol

2.7-Subjective Perception (this should also be included in the study design) or as a sub-section of the training protocol (ex., 2.6.1)

2.8-Dietary Protein Supplementation

2.9-Dietary Intake Recording

2.10-Statistical Analysis

Was the ultrasound method using a-mode or b-mode to capture the image? Additionally, the reliability of this analysis for muscle thickness should be noted.

Please be consistent with the identification of muscle groups. For example, line 219 identifies the vastus lateralis whereas line 190 refers to the M. Lateralis Femoris.

The training “load” is typically referred to as the weight of the resistance applied, whereas, the training volume is the product of the load and repetitions. Please modify the wording to reflect the accurate terminology.

Statistical analyses: I appreciate the simplicity of analyses as they are accurate and sufficient. However, the “unpaired” t-test or as otherwise called independent t-test should not be used to compare delta values. The comparison of delta inflates the chance of committing a type II error, when the actual raw data should be evaluated for significance and the mean differences between groups reported to supplement an effect size. Further, it would be beneficial to add in an analysis of effect size, such as, partial eta squared to enhance the defense for any significant effect identified.

Results: “Trends” should not be reported as significant. Generally, one would hope if a sample size increases, so would the strength of the effect, however, the data could easily go in a less-favorable direction. Therefore, noting a trend is not helpful, nor meaningful. (Figure 3. Squat)

The authors interchange the use of CP and COL. Please be cautious when referring to each and perhaps pick one for consistency throughout the paper.

Discussion, Lines 371-373: The discussion mentions an additional ANOVA comparing phase 1 to phase 2, in other comparisons these were grouped together as the intervention period rather than separate. If the authors did indeed split phase 1 and phase 2, then ALL comparisons should have included a 3-way multi-factorial ANOVA [phase (1 or 2) vs. time (pre or post) vs. condition group (COL or PLA)].

Comparisons of results throughout the discussion only compare to Zdzieblik et al., in which case the present study is trying to enhance the results of that study by investigating younger individuals. However, there are additional studies that are comparable and can be used to enhance the discussion to support the results identified in the present study.

Further, using the tissue that was extracted, there are methods to evaluate the passive tissue to identify if there is truly a change in that component rather than making assumptions (lines 378-380), considering the lack of change in contractile fibers. If the tissue is still available, it would be worth evaluating to strengthen the argument as stated in lines 383-384.

Lastly, it is important to differentiate the changes in type I vs type II fCSA. Changes were evaluated in type II which is in agreement with many other studies as well as the limited change in type I because they were performing resistance exercise that specifically riggers a response in type II. Please elaborate on this point in the discussion.

Author Response

Response to Reviewer 1 Comments

Dear reviewer,

First and foremost, I would like to thank you for reading and commenting on my manuscript. The comments are quite constructive and help me in this process to improve the manuscript. It is clearly recognizable that you have made a lot of effort. In the same way, I make every effort to answer your comments and to incorporate them into the manuscript.

Point 1: The statement, ”…it is unclear whether additional protein supplementation is able to enhance the beneficial effects of RET…” is not necessarily true. It has been repeatedly reported that Leucine, BCAAs, whey protein (isolate, hydrolysate, or concentrate), and plant-based proteins are indeed ergogenic in combination with RET. This statement should be modified or removed.

Response 1: This sentence was mainly related to the previously mentioned protein biosynthesis of passive structures. Here it is unclear whether an additional protein, especially for these structures, may favour the adaptations. The fact that this sentence can quite rightly cause confusion, it has been modified to: “However, it is unclear whether a specific protein supplementation is able to enhance the adaption of the mentioned passive tissue components” (lines: 45-46)

Point 2: The description of phase 1 and phase 2 to determine appropriate sample size is very confusing. It seems that phase 1 was exploratory and not sufficient, therefore additional subjects were recruited. This could be elaborated on briefly.

Response 2: We chose "adaptive design" as a study design (Chow, 2014 – Adaptive clinical trial design). Therefore, the first phase can be called exploratory. In this phase, we wanted to explore if it makes any sense to run this study on a larger cohort. After looking into the results of G-Power, the calculation of the group size of 28 (effect size: 0.78) was quite possible to achieve. Therefore, we started phase 2.

Point 3: Figure 1 indicated there was blood sampling, but only during post-testing. The description of blood sampling (section 2.8) was unclear if phase 1 blood sampling was day 3 of pre- or post-testing. However, only phase 2 was collected post-testing?

Response 3: Thank you for this hint. Section 2.8 has been modified: “Blood samples in phase 1 were collected on day 3 of post testing. Because there was no testing day 3 in phase 2, blood samples in phase 2 were collected at the end of the training period” (lines: 218-219).

Point 4: Why were muscle biopsies only taken during phase 1?

Response 4: The biopsies were used to gather muscle tissue that has been analysed for CSA to determine the muscle cell growth and possibly detect any differences in growth between our groups. But the analyses of the CSA in phase 1 was clear and unambiguously as no group differences in the adaptation were found (Anova: no interaction effect = p > 0.05).

Because this type of procedure always carries a small risk of infection and we have not expected any further gain of information by biopsies in phase 2, we decided against it. The decision was made by Prof. Dr. Petra Platen (Head of the corresponding department), Dr. Markus de Marées (medical doctor who performed the biopsies), my colleague Vanessa Oertzen-Hagemann and me in a detailed conversation. This is described and briefly modified in the results section “3.5 Muscle biopsy” (lines: 336 – 338)

Point 5: Was MViC conducted on the subject’s dominant leg? Or simple on all right or all left legs, based on how the dynamometer is set up? If the latter (ex, all right legs), then leg dominance should be noted as well as any recent (within 2 years) right leg injury (ACL, meniscus, strain, etc.).

Response 5: All MViC´s were performed on the right leg of each of our participants without a consideration of dominant or non-dominant legs. The first phase was completed in December 2016 and the second phase in July 2018. Therefore, it could be hard now to get the information of subject’s dominant leg or about injuries. Many already have other email addresses or mobile numbers. At the time of the study, all subjects assured me that they are injury free for at least 6 months.

Point 6: The methods should be listed in the order that they occurred, therefore, the sections after general study design should be re-ordered:

2.1-Subjects

2.2-Body Composition Testing

2.3-Ultrasound and Anthropometric Measurement

2.4-Strength Testing

 2.5-Muscle Biopsies and Venous Blood Sampling

 2.5.1 Immunohistochemical Stains of Skeletal Muscle-This is not a method involving contact with the subject

 2.6-Resistance Training Protocol

 2.7-Subjective Perception (this should also be included in the study design) or as a sub-section of the training protocol (ex., 2.6.1)

 2.8-Dietary Protein Supplementation

 2.9-Dietary Intake Recording

 2.10-Statistical Analysis

Response 6: This order really makes a lot more sense. I almost took that order. However, I still would like to list the biopsy sections at the end of the methods, because here we describe a methodology that only took place in phase 1. As well as in the methods, the biopsies are also listed in the results at the end to clarify this distinction between the whole cohort and only the subjects of phase 1. Although we have carefully considered this decision and for various reasons have renounced the biopsies in Phase 2, this poses a limitation and may cause confusion in the manuscript. Therefore, this distinction in the order is very important to me.

Point 7: Was the ultrasound method using a-mode or b-mode to capture the image? Additionally, the reliability of this analysis for muscle thickness should be noted.

Response 7: We used a b-mode system to capture the images (added here: line: 187). The reliability has been noted now with references to coefficients of variation and intraclass correlation coefficients (line: 360 – 363).

Point 8: Please be consistent with the identification of muscle groups. For example, line 219 identifies the vastus lateralis whereas line 190 refers to the M. Lateralis Femoris.

Response 8: The identification of the respective muscle groups was modified to:

m. vastus lateralis, m. vastus intermedius, m. rectus femoris.

Point 9: The training “load” is typically referred to as the weight of the resistance applied, whereas, the training volume is the product of the load and repetitions. Please modify the wording to reflect the accurate terminology.

Response 9: The description of the product of the weights and repetitions per training session was modified. Referred to the publication of Halson 2014 (Monitoring training load to understand fatigue in athletes) the terms of external and internal loads were implied. In this mentioned publication, the author defines and describes the training load the following way:

 “4 Internal versus External Load

When monitoring training load, the load units can be thought of as either external or internal. Traditionally, external load has been the foundation of most monitoring systems. External load is defined as the work completed by the athlete, measured independently of his or her internal characteristics [6]. An example of external load in road cycling would be the mean power output sustained for a given duration of time (i.e. 400 W for 30 min). While external load is important in understanding work completed and capabilities and capacities of the athlete, the internal load, or the relative physiological and psychological stress imposed is also critical in determining the training load and subsequent adaptation. As both external and internal loads have merit for understanding the athlete’s training load, a combination of both may be important for training monitoring.”

 The units or descriptors of training volume are noted here as “Time, intensity”

 Therefore, the internal load could be represented in our collected RPE values. This was modified in the manuscript too.

 So, what was originally meant with training load is now called external load in the manuscript. The external load is briefly defined in lines 159-162 with the reference to the mentioned publication.

 Point 10: Statistical analyses: I appreciate the simplicity of analyses as they are accurate and sufficient. However, the “unpaired” t-test or as otherwise called independent t-test should not be used to compare delta values. The comparison of delta inflates the chance of committing a type II error, when the actual raw data should be evaluated for significance and the mean differences between groups reported to supplement an effect size. Further, it would be beneficial to add in an analysis of effect size, such as, partial eta squared to enhance the defense for any significant effect identified.

 Response 10: In addition to the listing of all important values in Table 1, I wanted to visualize the difference between the groups in a simplified way. Therefore, the delta values were used. And just because I chose this type of visualization, I had to adjust my statistics too. But I totally agree with your comment regarding the delta values and its statistically limitation. I have decided to remove the figure to avoid exactly this problem. Or would you say there is a way to represent the delta values for a better visualisation of the differences between groups but still use the “right” statistic? For example: Show delta values but with Anova results above the bars?

 Partial Eta squared ( ) was implemented to the statistics section (lines: 266 – 268). The effect size of the significant interaction effects (group * time) were calculated and implemented in the results section for FFM (line: 296), BFM (line: 299) and SQ (line: 309).

 Point 11: Results: “Trends” should not be reported as significant. Generally, one would hope if a sample size increases, so would the strength of the effect, however, the data could easily go in a less-favorable direction. Therefore, noting a trend is not helpful, nor meaningful. (Figure 3. Squat)

 Response 11: This was corrected to “§ = trend for an interaction effect (time * group, p = 0.054). (line: 315).

 Point 12: The authors interchange the use of CP and COL. Please be cautious when referring to each and perhaps pick one for consistency throughout the paper.

 Response 12: While CP (collagen peptides) is the supplement and used to describe the collagen supplementing groups of other studies (line 357: of their CP supplementing group compared to…), COL (collagen peptide supplementation group) is the specific group in our study. Therefore, one of the two terms cannot be waived. If these terms lead to confusion, I may also add an explanatory sentence in the methods.

 Point 13: Discussion, Lines 371-373: The discussion mentions an additional ANOVA comparing phase 1 to phase 2, in other comparisons these were grouped together as the intervention period rather than separate. If the authors did indeed split phase 1 and phase 2, then ALL comparisons should have included a 3-way multi-factorial ANOVA [phase (1 or 2) vs. time (pre or post) vs. condition group (COL or PLA)].

 Response 13: A comparison of phase 1 and phase 2 is not usual to be considered in an adaptive study design method. We did not expect differences between phase 1 and phase 2 due to the identical methods, investigators and characteristics of participants. An additional 3-way multi-factorial ANOVA was therefore not planned and would not reveal any new result to our main question if collagen peptides have an influence on strength or body composition. Because no differences were found in the mentioned two-way ANOVA (phase * group) between groups and phases in FFM and muscle thickness via ultrasound, we assume that there would also be no difference in CSA adaptation between phases. The mentioned lines should just strengthen the assumption that the adaptation of the CSA is just as compatible as all other parameters of the body composition between phase 1 and 2, although there is no data about CSA in phase 2. This additional ANOVA should only strengthen this assumption and reduce the limitation a bit (biopsies just in phase 1).

 Point 14: Comparisons of results throughout the discussion only compare to Zdzieblik et al., in which case the present study is trying to enhance the results of that study by investigating younger individuals. However, there are additional studies that are comparable and can be used to enhance the discussion to support the results identified in the present study.

 Response 14: The discussion has been extended.

Point 15: Further, using the tissue that was extracted, there are methods to evaluate the passive tissue to identify if there is truly a change in that component rather than making assumptions (lines 378-380), considering the lack of change in contractile fibers. If the tissue is still available, it would be worth evaluating to strengthen the argument as stated in lines 383-384.

 Response 15: Fortunately, there is some tissue left in the labs of the sports University of cologne. The times in the labs are very limited and that’s why we just checked the CSA to get a first idea about changes in contractile muscle tissue mass. It is planned to look at our remaining 7 µm thick serial sections via Trichrom-coloring method after Masson Goldner. This will take us a while but it is planned after the summer. If you have an idea about more adequate methods, I would very much appreciate receiving this information from you. However, only serial sections are left and no coherent piece of tissue.

 Point 16: Lastly, it is important to differentiate the changes in type I vs type II fCSA. Changes were evaluated in type II which is in agreement with many other studies as well as the limited change in type I because they were performing resistance exercise that specifically riggers a response in type II. Please elaborate on this point in the discussion.

 Response 16: A brief paragraph was added with a reference that strengthens our results as seen in differences in fCSA adaptation between the respective fiber types (line: 368-370).

Reviewer 2 Report

Participants of the study have to be clarified, there is contradictory information in methods section.

Provide exact p values

Author Response

Response to Reviewer 2 Comments

 Dear reviewer,

First and foremost, I would like to thank you for reading and commenting on my article. The comments are quite constructive and help me in this process to improve the manuscript.

Point 1: Participants of the study have to be clarified, there is contradictory information in methods section.

Response 1: The number of participants differs in the method section. The reason is the use of the “adaptive design” method. Therefore, two phases were used to increase the power of the study. Possible differences between the groups were collected in phase 1 with 25 participants and after, another 40 participants have been recruited in phase 2. Merged together (phase 1 + phase 2), 68 subjects started the study but only 57 completed it successfully. In the same section, the biopsies were described. This is separated because this shows the only method that just happened in phase 1. The biopsies were used to gather muscle tissue that has been analysed for CSA to determine the muscle cell growth and possibly detect any differences in growth between our groups. But the analyses of the CSA in phase 1 was clear and unambiguously as no group differences in the adaptation were found (Anova: no interaction effect = p > 0.05).

Because this type of procedure always carries a small risk of infection and we have not expected any further gain of information by biopsies in phase 2, we decided against it. The decision was made by Prof. Dr. Petra Platen (Head of the corresponding department), Dr. Markus de Marées (medical doctor who performed the biopsies), my colleague Vanessa Oertzen-Hagemann and me in a detailed conversation. This is described and briefly modified in the results section “3.5 Muscle biopsy” (lines: 348 – 350).

 Therefore, the participants of the study are merged from phase 1 to phase 2 but separated in terms of the results of the muscle biopsies.

 Point 2: Provide exact p values

 Response 2: The exact p values have been inserted. The respective sections and spots in the text are marked in yellow.

 Reviewer 3 Report

The manuscript by Kirmse et al. entitled "Prolonged collagen peptide supplementation and RET affects body composition in recreationally active men" determined the effects of collagen supplementation for 12 weeks in combination with resistance exercise training on body composition, strength, and muscle fiber cross sectional area in recreationally active men.

It is stated that the aim of this study (lines 74-75) is to re-examine the results of Zdzieblik et al. in a younger human cohort. Would part of the novelty of this study be that it re-examined and extended, e.g., muscle biopsies, the results found by others?   

Has the dietary supplement, viz., BodybalanceTM, been characterized? That is, what is the hydroxyproline content, the amino acid composition, the relative purity, etc.? Are there any adulterants?

Author Response

Response to Reviewer 3 Comments

 Dear reviewer,

First and foremost, I would like to thank you for reading and commenting on my article. The comments are quite constructive and help me in this process to improve the manuscript.

Point 1: It is stated that the aim of this study (lines 74-75) is to re-examine the results of Zdzieblik et al. in a younger human cohort. Would part of the novelty of this study be that it re-examined and extended, e.g., muscle biopsies, the results found by others?

Response 1: Indeed, the novelty of this study is the re-examination in a younger cohort and an additional extension of the methodology in the form of muscle biopsies. In the mentioned lines it was pointed out here: “To gather more information about the composition and possible adaptations of the FFM, invasive methods were incorporated to quantify isolated contractile muscle fiber hypertrophy in addition to strength and body composition analysis.” Instead of the term “invasive methods” the term “muscle biopsies” were now used to make this clearer and the structure of this whole section (lines 74-77) has been adapted.

Point 2: Has the dietary supplement, viz., BodybalanceTM, been characterized? That is, what is the hydroxyproline content, the amino acid composition, the relative purity, etc.? Are there any adulterants?

Response 2: The collagen supplement has already been described in the study of Zdzieblik et al. 2015. Due to the length of the current article, a detailed repetition has been avoided. A reference to the study by Zdzieblik et al. 2015, that has been missing so far, has now been added to the passage in the methods "Dietary protein supplementation" next to "BodybalanceTm" (line: 169).

The amino acid composition of the collagen peptides therefore is:

Amino acid        Weight (%)  Mol (%)

Hydroxyproline  11,3              9,6

Aspartic acid       5,8               4,8

Serine                  3,2               3,4

Glutamic acid 1   0,1               7,5

Glycine               22,1             32,3

Histidine             1,2               0,8

Arginine             7,8               5,0

Threonine           1,8               1,7

Alanine               8,5               10,5

Proline               12,3             11,8

Tyrosine             0,9               0,5

Hydroxylysine   1,7               1,2

Valine                 2,4               2,3

Methionine         0,9               0,9

Lysine                3,8               2,9

Isoleucine           1,3               1,1

Leucine              2,7               2,3

Phenylalanine     2,1               1,4

 The amino acid composition was determined by amino acid analysis as described in Pharm. Eu. 2.2.56 (Version 8). The proteins were hydrolysed for 24 h to their individual amino acid constituents in the presence of 6 n HCl and 0.1 % phenol at 110 °C. The amide links in the side chains of glutamine and asparagine are hydrolyzed to form glutamic acid and aspartic acid. Following the hydrolysis, the amino acids are covalently labelled with 6 – aminoquinolyl-N-hydroxysuccinimidyl carbamate (AQC; AccQ-Flour reagent, Waters Inc.) using aprecolumn derivatisation technique. L-2 Aminobutyric acid (AAbA) with a final concentration of 10 pmol/μl was used as internal standard. The derivatives are separated by C18 reversed-phase HPLC and quantified by fluorescence detection.

The protein content of the dry substance was tested via Kjeldahl (N x 5.55) Method and the result is ≥ 96.0 %. Therefore, the purity is very high.

In addition, the manufacturers deny any adulterant

Round  2

Reviewer 1 Report

Thank you for addressing my concerns. I feel you did an adequate job at addressing all concerns and modification of the manuscript has improved the integrity of the results.